# Interictal discharges spread along local recurrent networks between tubers and surrounding cortex

Stasa Tumpa[1,2], Rachel Thornton[3,4], Martin M. Tisdall[4,5] (iD), Torsten Baldeweg[4], Karl J. Friston[1] (iD) and Richard E. Rosch[1,6] (iD)

[1] *Wellcome Trust Centre for Neuroimaging, UCL Queen Square Institute of Neurology, University College London, London, UK*
[2] *School of Clinical Medicine, University of Cambridge, Cambridge, UK*
[3] *Department of Clinical Neurophysiology, Cambridge University Hospitals NHS Foundation Trust, Cambridge, UK*
[4] *UCL Great Ormond Street Institute of Child Health, University College London, London, UK*
[5] *Department of Neurosurgery, Great Ormond Street Hospital for Children NHS Foundation Trust, London, UK*
[6] *Department for Basic and Clinical Neuroscience, Institute of Psychiatry, Psychology and Neuroscience, King's College London, London, UK*

Handling Editors: Richard Carson & Bettina Schwab

The peer review history is available in the Supporting Information section of this article (https://doi.org/10.1113/JP288141#support-information-section).

**Abstract figure legend** Interictal epileptiform discharges (IEDs) indicate irritable cortex and may arise from brain lesions such as cortical tubers in tuberous sclerosis. Here, we identify IEDs in intracranial recordings in patients undergoing evaluation for epilepsy surgery, and demonstrate that they behave like a travelling wave arising from the tuber core. Using a neural mass model, we then demonstrate that recurrently coupled networks allow for the fast spread of IEDs along the tuber and perituberal cortical areas (left). Simulating a reduction of inhibitory connectivity, the fully parameterised network model then indicates that these recurrent networks, but not other types of network architectures, are capable of generating interictal epileptiform discharges, as well as seizure-like dynamics (right).

This article was first published as a preprint. Tumpa S, Thornton R, Tisdall M, Baldeweg T, Friston KJ, Rosch RE. Interictal discharges spread along local recurrent networks between tubers and surrounding cortex. bioRxiv. https://doi.org/10.1101/691170

The Journal of Physiology

**Abstract**  The presence of interictal epileptiform discharges on EEG may indicate increased epileptic seizure risk. In highly epileptogenic lesions, such as cortical tubers in tuberous sclerosis, these discharges can be recorded with intracranial stereotactic EEG as part of the evaluation for epilepsy surgery. Yet the network mechanisms that underwrite the generation and spread of these discharges remain poorly understood. Here, we investigate the dynamics of interictal epileptiform discharges using a combination of quantitative analysis of invasive EEG recordings and mesoscale neural mass modelling of cortical dynamics. We first characterise spatially organised local dynamics of discharges recorded from 36 separate tubers in eight patients with tuberous sclerosis. We characterise these dynamics with a set of competing explanatory network models using dynamic causal modelling. Bayesian model comparison of plausible network architectures suggests that the recurrent coupling between neuronal populations within, as well as adjacent to, the tuber core explains the travelling wave dynamics observed in these patient recordings. Our results indicate that tuber cores are the spatial sources of interictal discharges that behave like travelling waves with dynamics most probably explained by locally recurrent tuber–perituberal networks. This view integrates competing theories regarding the pathological organisation of epileptic foci and surrounding cortex in patients with tuberous sclerosis by through coupled oscillator dynamics. This recurrent coupling can explain the spread of ictal dynamics and also provide an explanation interictal discharge spread. In the future, we will explore the possible implications of our findings for epilepsy surgery approaches in tuberous sclerosis.

(Received 19 November 2024; accepted after revision 21 February 2025; first published online 15 March 2025)

**Corresponding author** R. E. Rosch: Maurice Wohl Clinical Neuroscience Institute, King's College London, 5 Cutcombe Rd, Brixton, London SE5 9RT, UK.    Email: richard.rosch@kcl.ac.uk

**Key points**

- Interictal epileptiform discharges (IEDs) are abnormal electrical patterns observed in the brains of people with epilepsy and may indicate seizure risk.
- In tuberous sclerosis, a condition causing epileptic lesions called cortical tubers, IEDs spread from the tuber core to surrounding brain tissue, forming travelling waves.
- This study used invasive EEG recordings and mathematical models to identify that recurrent connections between the tuber core and its surroundings explain this wave-like spread.
- Further *in silico* simulations demonstrate that this recurrent network architecture supports both interictal discharges and seizure-like dynamics under different levels of local inhibition

# Introduction

Patients with epilepsy experience recurrent seizures caused by abnormal, hypersynchronous brain activity (Fisher et al., 2005). Most patients achieve seizure control with anti-epileptic drugs, but around one-third continue to have seizures despite treatment (Berg & Rychlik, 2015). For these patients, neurosurgical removal of the putative epileptogenic zone has emerged as an efficacious treatment (Duncan et al., 2016; Rosenow & Lüders, 2001). Some patients have seizures after surgery, suggesting that their epilepsy did not arise from focal abnormal activity

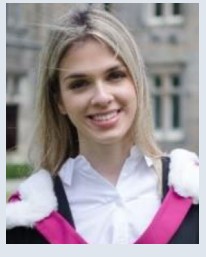

**Stasa Tumpa** is a resident doctor (Neurosurgery FY2) in Addenbrooke's Hospital, Cambridge. She holds a BSc (Hons) and MSc in Neurosciences from St Andrews and University College London, respectively, completing internships at Johns Hopkins and Purdue University. She completed her clinical training at the University of Cambridge, where she developed further research interests in global neurosurgery, scientific dissemination and mixed methods neurosurgical research.

alone. Indeed, epilepsy is now considered a network disorder with emergent dynamics in an abnormally coupled network (da Silva et al., 2012).

Tuberous sclerosis complex (TSC) is a genetic multisystem disorder, and a leading cause of epilepsy and autism (Osborne et al., 1991). TSC is associated with lesions in many organs, including brain, heart, skin, kidneys and lungs (Curatolo et al., 2008). Intracranial lesions are predominantly cortical tubers, which are highly epileptogenic and characterised by aberrant neuronal coupling in both tuber and surrounding tissue (perituberal cortex) (Ruppe et al., 2014). Tuber core and perituberal cortex, or both, have been purported as possible seizure onset zones in patients with TSC (Kannan et al., 2016; Ma et al., 2012; Major et al., 2009). Identifying the local network mechanisms underlying seizures in these patients is a topic under active debate (Gupta, 2017) with implications for surgical treatment, the outcomes of which are currently still variable (Fallah et al., 2015).

Epilepsy surgery in patients with TSC usually involves intracranial EEG evaluation of seizure networks (Bollo et al., 2008) because scalp EEG and imaging alone can be poorly predictive of outcomes (Fallah et al., 2013). Recording intracranial EEG from stereotactically implanted depth electrodes (SEEG) allows sampling from multiple candidate tubers, but the widespread abnormalities, varied seizure propagation pathways and spatial sampling limitations make evaluating recordings challenging. Extensive resection of the candidate tubers and surrounding cortex are commonly performed even after detailed investigation (Wang et al., 2014). Thus, improved understanding of tuber-related epileptogenicity may help develop more restrictive resections with better surgical outcomes.

The seizure onset zone, namely the target of neurosurgical intervention, is defined by the ictal activity. Interictal epileptiform discharges (IEDs) can provide complementary localising information (Murakami et al., 2016; Staley & Dudek, 2006), indicate dynamic changes in seizure risk (Baud et al., 2018) and are associated with synaptic connectivity changes potentially contributing to epileptogenesis (Staley et al., 2005). IEDs with the same features are seen across widely spaced electrodes with latencies of tens of milliseconds, suggesting rapid propagation to distant cerebral structures (Alarcon et al., 1997; Emerson et al., 1995; Wendling et al., 2012).

This propagation speed is in stark contrast to epileptic seizures themselves, with ictal recruitment often taking several seconds. Yet the same networks support slow ictal and fast interictal propagation (Proix et al., 2018). Spatiotemporal dynamics of IEDs may thus reveal additional insights into the functional organisation of cortical areas involved in seizure generation and spread (Bettus et al., 2008).

Travelling wave dynamics can emerge from a number of different network architectures, which, at their most fundamental, represent two competing possible models: (1) a fixed sequential delay emerging from message passing in one direction along network coupling or (2) an almost arbitrary delay emerging from the specific coupling dynamics of a recurrently coupled network (Ermentrout & Kleinfeld, 2001). Here, we quantify and model the dynamics of individual IEDs, aiming to infer a neurobiologically plausible connectivity structure. We identify systematic IED delays between neighbouring SEEG channels. We then fit network architectures that recapitulate two possible propagation modes: (1) propagation along recurrent coupled oscillators and (2) propagation of pulses in an excitable network (Muller et al., 2018; Proix et al., 2018). We hypothesised that the fast spread of interictal discharges may reveal existing recurrent coupling in local epileptogenic networks [i.e. provide evidence for model architecture (2) above], which can ultimately support both ictal and interictal discharges.

To compare different explanatory networks, we use dynamic causal modelling (DCM), a framework that allows quantitative comparison of candidate explanatory models (Kiebel et al., 2008; Moran et al., 2008; Moran et al., 2011) using neural mass models of neuronal population dynamics [e.g. the canonical microcircuit (Bastos et al., 2012) used here]. Furthermore, recent advances in the efficient estimation across large model spaces (Friston et al., 2018) allow us to compare model architectures at the level of individual tubers.

## Methods

### Ethical approval

Anonymised data were collected retrospectively from recordings performed according to standard clinical practice for a clinical indication. The study did not influence clinical decision making, and the approach to recording intracranial EEG was not altered by the research question; all care for the patients was provided according to standard clinical practice. The retrospective use of anonymised clinical data was approved by the UK Health Regulary Authority (HRA, IRAS ID 229 772) and the Great Ormond Street Hospital/UCL Great Ormond Street Institute of Child Health Joint Research Office (Project ID 17NP05). HRA approval specifically waived consent for this retrospective analysis of anonymised data.

### Patient identification

We retrospectively considered all patients admitted to a tertiary paediatric epilepsy surgery centre between January 2016 and June 2018 and included them if they

**Table 1. Patient details.**

| Patient | Age at epilepsy onset (years) | Age at SEEG recording (years) | Number of SEEG Electrodes |
|---|---|---|---|
| TS_01 | 1–3 | 6–8 | 6 |
| TS_02 | 0–1 | 3–5 | 8 |
| TS_03 | 1–3 | 3–5 | 16 |
| TS_04 | 1–3 | 3–5 | 9 |
| TS_05 | 0–1 | 9–11 | 12 |
| TS_06 | 0–1 | 9–11 | 9 |
| TS_07 | 0–1 | 12–15 | 5 |
| TS_08 | 0–1 | 6–8 | 7 |

(1) had a diagnosis of TSC and (2) underwent presurgical evaluation with SEEG recording for pharmacoresistant epilepsy. All eight patients had focal seizures and active focal epilepsy with onset in infancy or early childhood. Patient details are summarised in Table 1.

### Intracranial EEG acquisition, selection and pre-processing

**SEEG recording.** SEEG electrodes were placed based on multidisciplinary SEEG planning involving neurologists, neurophysiologists and neurosurgeons according to clinical need, informed by presurgical imaging, seizure semiology, and ictal and interictal scalp EEG. SEEG depth electrodes (DIXI Medical, Marchaux-Chaudefontaine, France) were inserted under stereotactic guidance using the neuromate® (Renishaw, Wotton-under-Edge, UK) robot system (programmed with patient specific pre-operative imaging) into various structures, including multiple tubers in each child (Sharma et al., 2019). Most electrodes had one or more contacts in the tuber core, with additional contacts superficial, and/or deep to the core (i.e. in superficial grey matter and deep white matter respectively) (Fig. 1*A*). SEEG was recorded continuously for up to 5 days to capture seizures for clinical interpretation. All SEEG recordings were recorded using a NeuroWorks system (Natus, Middleton, WI, USA) at a sampling rate of 1 kHz, with a white matter contact remote from regions involved in the generation of seizures used for reference. Quantitative analysis of delays was performed on bipolar montage.

**Classification of SEEG contact position.** Pre-implantation magnetic resonance imaging fluid-attenuated inversion recovery (FLAIR), T1 and T2-weighted images (WI) were co-registered with post-implantation computed tomography to visually determine the position of SEEG contacts in relation to cortical tubers, perituberal grey

and white matter. In young children, tubers appear hypointense on the T1-WI image and hyperintense on T2-WI and FLAIR images (Grajkowska et al., 2010). Although, in some patients, generalised brain volume loss was observed, most affected gyri were enlarged with blurring of the grey–white junction (Jurkiewicz et al., 2006; Weisenfeld et al., 2013). Individual contacts were classified in terms of their position to the tuber core on an integer scale centred around 0 (tuber core position) by consensus of two raters (MT and ST), including a neurosurgeon with clinical expertise in visual identification of TSC features on imaging.

**Time series extraction.** In total, 15 one minute segments of interictal extra-operative SEEG recordings were selected for each patient, providing sufficient IED counts for subsequent analysis. Data were extracted as individual, 60 s intervals with a random selection procedure implemented by drawing random onset times using a custom MATLAB (MathWorks, Natick, MA, USA) routine. Segments containing visible artefact or clinically labelled seizure activity were excluded. This approach yielded a broad coverage of behavioural states (e.g. awake and asleep) and provided a median of ∼40 spikes per SEEG electrode included in the analysis. Band pass filter (0.5–120 Hz, zero phase) and notch filter (50 Hz) were applied for visual inspection on the clinical Natus Database system. The signal was re-montaged to bipolar montages between adjacent contacts on the same SEEG electrode and exported in EDF+ format for further processing using custom MATLAB code, which is available online (https://github.com/roschkoenig/travelling_spikes).

**Spike detection.** A custom spike detection algorithm [adapted from SPKDT v1.0.4 (Barkmeier et al., 2012) to accommodate the values below] was run to identify individual interictal epileptiform discharges. Criteria for spike detection were: (1) peak amplitude >4 SD from the mean; (2) ascending and descending absolute slope values of >7 µV/msl and (3) total width of spike <20 ms. We then grouped spikes recorded from different contacts of the same SEEG electrode and detected groups of spikes that co-occurred within 200 ms of each other in time across channels on the same SEEG electrode. Only spike groups with at least two spikes within this time window were considered for further analysis. Delay was then estimated between all available channels centred on the timing of the detected spikes, which means that, in the bipolar montage (where spikes that appear similar across neighbouring contacts may be detected only as small amplitude deflections), delays can still be identified across the full range of the SEEG electrode contacts.

## Sensor space analysis

In the first instance, we analysed data features of the recorded SEEG signal to determine whether the spatiotemporal distribution of interictal spikes was consistent with a travelling-wave spread and tested these using Bayesian statistical criteria.

**Delay estimation.** We estimated the temporal delay between clusters of spikes spanning several channels of the same SEEG electrode by calculating the cross-correlation between the signal and estimating the delay between neighbouring channels at which cross-correlation is maximal. Because the fast (i.e. 'spike', typically gamma-frequency range) and the slow (i.e. 'wave', typically delta range) components of a spike and wave discharge may exhibit different dynamics, we filtered the continuous EEG signal into two frequency bands: Low frequencies (<13 Hz, delta to alpha) and high frequency (13–120 Hz, beta to gamma). After filtering, delay was

assessed within a 200 ms window, centred on the earliest detected spike for both frequency bands independently.

**Bayesian statistics on spatiotemporal spike distribution patterns.** To identify the spatiotemporal organisation of interictal discharges, we compared different explanatory models linking the relative time of spike detection to the position of the channel in relation to the tuber core. We proposed three different possible (linear) models that could explain the spatiotemporal organisation [where $\mathbf{p}$ is the vector of positions $(p_1, p_2, …, p_n)$] a spike was detected at, and $\mathbf{t}$ is the corresponding vector of time points of detections $(t_1, t_2, …, t_n)$. Vector $\mathbf{p}$ is encoded relative to the tuber core, with deep white matter represented by the most negative value, the core position represented by zero, and overlying grey matter represented by positive values):

*Uniform.* If there is no spatiotemporal pattern, time of detection should be independent of channel position, modelled through a simple constant function (where $b_{uniform}$ defines a relationship where $\mathbf{t}$ is independent of

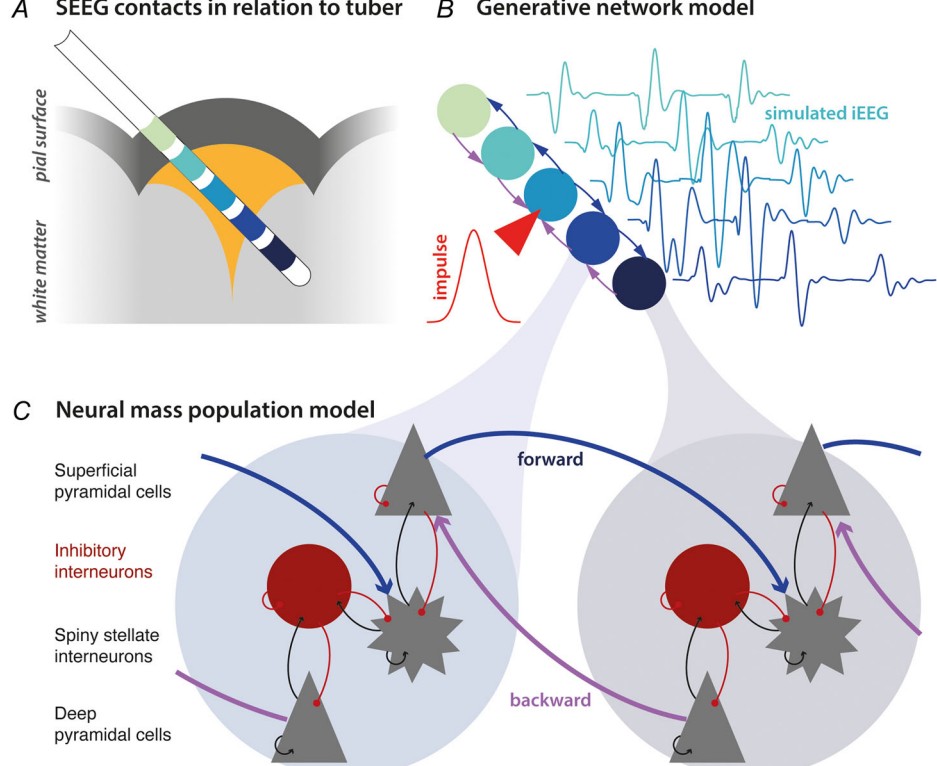

**Figure 1. Dynamic causal modelling**
*A*, intracranial interictal recordings were acquired with stereotactically implanted depth electrodes (SEEG), and visually classified in terms of their relation to the tuber core. *B*, dynamic causal modelling allows the testing of network models of dynamics that generate SEEG responses to endogenous fluctuations or spikes, in this case modelled as an external impulse that prompts a distributed pattern of interictal epileptiform discharges. We inverted such models to best explain interictal epileptiform discharges. *C*, each node or 'source' in the model comprises coupled neural mass models of four populations, organised into two oscillator pairs (superficial pyramidal cells and spiny stellate cells; deep pyramidal cells and inhibitory interneurons). [Colour figure can be viewed at wileyonlinelibrary.com]

**p**, thus on average constant with respective to changes in position **p**):

$$\mathbf{t} = b_{uniform} \tag{1}$$

*Depth to surface.* If interictal discharges spread through the cortical tuber along the grey–white matter axis, we would expect a linear relationship between channel position and time of spike detection as follows (where $a_{depth}$ defines the slope and $b_{depth}$ is the intercept of a linear relationship between **t** and **p**):

$$\mathbf{t} = a_{depth} * \mathbf{p} + b_{depth} \tag{2}$$

*Core to periphery.* This model encodes a spread of interictal discharges between core and surrounding tissue through a relationship of absolute distance from core, and time of detection (where $a_{depth}$ defines the slope and $b_{depth}$ is the intercept of a linear relationship between **t** and the absolute distance from the tuber core encoded by $|\mathbf{p}|$):

$$\mathbf{t} = a_{core} * |\mathbf{p}| + b_{core} \tag{3}$$

We fit each of these simple linear models to the spike groups detected in the earlier steps and compared models using the Bayesian information criterion (BIC) for each of these models. The BIC is defined as $BIC = \log(n)k - 2\log(\hat{L})$, where $n$ is the sample size, $k$ the number of free parameters estimated by the model fit and $\hat{L}$ is the maximised likelihood function of the model. This measure allows a comparison of the relative evidence for models with different numbers of parameters because it integrates measures of model accuracy (i.e. the maximal likelihood function) at the same time as penalising models with large numbers of free parameters (i.e. large $k$). Lower BIC values imply either fewer explanatory variables, better fit or both (Schwarz, 1978), and the model with the lowest BIC has the best explanatory power for the observed data. When $n$ is large, a difference in BIC ($\Delta BIC$) is proportional to the Bayes factor, which can be used for Bayesian model comparison, and a value of $>10$ is considered very strong evidence (Jones et al., 2001). The model comparison is implemented in Matlab with custom scripts accessible (https://github.com/roschkoenig/Travelling_Spikes) and specification of models contained in the `ts_delaystats.m` script.

## DCM

**DCM framework.** Macroscopic neuronal dynamics as measured by SEEG can be generated by mesoscale neural mass models that describe the average behaviour of neuronal populations and their coupling, as pioneered by Wilson & Cowan (1973). Here, we use DCM (Friston et al., 2003) to compare different models of local network coupling underlying the dynamics of inter-

ictal discharges. We use the canonical microcircuit model (Bastos et al., 2012), which is widely used in DCM, including work in epilepsy and other neurological abnormalities (Auksztulewicz & Friston, 2015; Bastos et al., 2015; Litvak et al., 2015; Papadopoulou et al., 2015; Pinotsis et al., 2017). We perform this analysis in a fixed-point regime, where IEDs are modelled as transient, provoked perturbations that elicit damped oscillatory dynamics around the stable fixed point attractor of the specified model.

This model comprises four coupled neural masses, organised in two excitatory/inhibitory oscillator pairs, and parameterised by coupling strength parameters and time constants according to the flow equations:

$$\dot{i}_k = i_k \tag{4}$$

$$\dot{i}_k = \frac{H_k}{\tau_k} \sum d_{kl} \sigma(v_l) \tag{5}$$

Here, $v_k$ is membrane voltage, $i_k$ is the average current, $H_k$ is the maximum postsynaptic potential and $\tau_k$ is the time constant of population $k$. Changes in current and voltage are induced by input from other populations $l$, weighted by their coupling to population $k$ via coupling parameter $d_{l-k}$. The input is passed through a sigmoid function representing population variance $\sigma$. Coupling in our model is formally separated into extrinsic connectivity (i.e. the $A$ matrix) and a set of local coupling parameters $\gamma$. For each region, DCM will update all external coupling parameters, selected $\gamma$ parameters that maximally affect neuronal population responses, population specific time constants and the sigmoid parameter (Table 2). Full specifications of this model are provided elsewhere (Moran et al., 2013).

DCM assumes that the activity in one source is evoked by the activity in another (David & Friston, 2003) and DCM for EEG uses the above mentioned neural mass model to explain the source activity and causal interactions (Garrido et al., 2008). Rather than estimating source activity at isolated points in time, it models source activity over time, accounting for the interacting inhibitory and excitatory populations of neurons. DCM uses variational Bayesian model inversions to infer the parameters that best explain the source activity and provides a free energy approximation to the model evidence. This can be used for Bayesian model comparison (compare the use of BIC when comparing data descriptor models above).

The principal aim of our DCM analysis was to compare different models that can explain the IED data features observed. Specifically, we were interested in identifying systematic features of the coupling between the tuber core and surrounding cortex (i.e. the perituberal network). We therefore constructed two competing network models:

**Table 2. Group level parameter averages.**

| Parameter | Prior value | Posterior estimate (Bayesian average) |
|---|---|---|
| $A$ forward | 1.00 | 1.55 |
| $A$ backward | 1.00 | 1.97 |
| $\gamma_{sp-sp}$ | 1.00 | 1.18 |
| $\gamma_{sp-ss}$ | 1.00 | 1.04 |
| $\gamma_{ii-ss}$ | 1.00 | 1.11 |
| $\tau_{ss}$ | 1.22 | 2.34 |
| $\tau_{sp}$ | 0.90 | 3.30 |
| $\tau_{dp}$ | 1.22 | 1.68 |
| $\tau_{ii}$ | 0.90 | 3.10 |
| $\sigma$ | 1.00 | 1.10 |

Coupling parameters for the networks of neural masses fitted to individual IEDs are shown as the group level Bayesin average. Both prior values and posterior estimates are shown for between region coupling ($A$), local inhibition or population gain $\gamma$, population time constants $\tau$ and population variance $\sigma$. Indices refer to neuronal populations: ss, spiny stellate cells; sp, superficial pyramidal cells; dp, deep pyramidal cells; ii, inhibitory interneurons.

each competing model comprised multiple sources (one per SEEG contact), arranged along an axis from deep white matter to the tuber core and then to the peri-tuberal grey matter (i.e. arranged from depth to surface). They differed only in the coupling between sources: one model ('forward only') had connections from the tuber core leading 'outwards' towards both deep white matter and perituberal grey matter; the second model ('recurrent coupling') had connections from and to each pair of neighbouring sources.

In standard event-related potential applications of DCM, aiming to improve signal to noise ratio, averages of many instances of a cortical potential are used as the data features. In our case, we found that averaging IEDs within an electrode could blur the shape of the discharges, as a result of small variations in their morphology across the sample. We therefore opted to model individual IEDs (rather than averages). In a trade-off between computational load of the analysis and capturing the maximum variability present in the data set, we used a data-driven clustering approach that helped us select a set of three IEDs per patient that were representative of the whole dataset as follows (this can be regarded as a form of robust averaging within similar response classes). (1) We performed a principal component analysis of the multichannel recordings of all individual IEDs in a single SEEG electrode and retained the components necessary to explain 90% of the variance in the dataset. (2) In this low dimensional representation of the data, we performed *k*-means clustering, aiming to identify groups of IEDs that were morphologically similar at the same time as

being as distinct as possible from other groups. (3) Having identified the centroids of these clusters, we then selected the IED closest (in principal component space) to the centroid as representative of the group for subsequent inversion. A worked example is provided in Fig. 2.

This resulted in three representative IEDs for each tuber. We then fitted our DCMs to these data for each SEEG electrode using Bayesian model inversion. Data for the inversion were limited to a window of –100 ms to 250 ms around the peak of the first spike detected within a temporal Hanning window. We assumed a Gaussian input function with prior expectations of 0 ms onset and 16 ms width (to model endogenous discharge events). The exact timing and width was optimised as part of the model inversion.

**Bayesian model reduction and model comparison.** DCM yields two main results. (1) The inversion of a model using the Bayesian approach implemented in DCM (Friston et al., 2007) provides a quantitative estimate of the model evidence, which is approximated by the (negative) free energy [i.e. $F \approx -\ln p(y|m)$, where $F$ is the free energy, $y$ is the data, $m$ is the model and $\ln p(y|m)$ is the log evidence]. The difference in free energy between competing models corresponds to the logarithm of the corresponding Bayes factor (Baele et al., 2013; Lin & Yin, 2015), used for model comparison. (2) At the same time, the inversion yields parameter estimates. As part of the Bayesian inversion, these estimates are not just single values, but parameter densities (i.e. they encode the expected value of the parameter as well as a measure of uncertainty around that value). Thus, parameter values that are well-constrained by the data have very narrow densities.

For the analysis here, we focus on comparison between model architectures, rather than inference on different parameters; i.e. hypothesis testing is performed using Bayesian model comparison. We do not assume that different tubers necessarily have the same perituberal network architecture, and therefore allow for random effects in our analysis. A full discussion of Bayesian random effects model comparison is provide in Penny et al. (2010). Traditionally, this would be achieved by fitting the separate model architectures individually to each dataset and then comparing the ensuing free energies. However, the free energy of each model can be evaluated more efficiently through Bayesian model reduction (Friston et al., 2018). Instead of fitting each candidate model separately, we invert only the 'full' model for each dataset (i.e. a model containing all plausible free parameters). We can then efficiently estimate the model evidence for reduced or 'nested models' that retain subsets of the original parameters by 'switching off' some of the parameters (with shrinkage priors). A detailed discussion

of this approach has previously been described (Friston et al., 2015).

**Model specification.** We specified our candidate network models of local coupling along two main dimensions:

(1) *Recurrent vs. outward only coupling*: Previous mathematical models of travelling wave propagation and ictal wave propagation suggested different modes of wave propagation (Ermentrout & Kleinfeld, 2001; Muller et al., 2018; Smith et al., 2016). (1) Travelling waves propagating through excitable neural media. Here, the wave originates at a single oscillator (i.e. a pacemaker), directly exciting the neighbouring regions of the cortex through a progression of increasing time delays. (2) Message passing along a recurrently coupled network. Crucially, in this model architecture, all local oscillators are potential sources for rhythmic outputs.

(2) *Superficial, core or deep source*: In the neural mass formulation, an impulse will trigger the subsequent spread of activity. Here, we model this as Gaussian input into key positions of the network established above; specifically, the perituberal grey (most superficial), the tuber core itself (core) or the deep white matter (deepest) nodes.

This gives us a model space of $2 \times 3 = 6$ models for each individual SEEG electrode. We then evaluated group-level evidence for each network architecture using a (variational) free energy bound on Bayesian model evidence (see below).

**Simulations.** Finally, we took a representative example of a full network fitted in the previous step and simulated

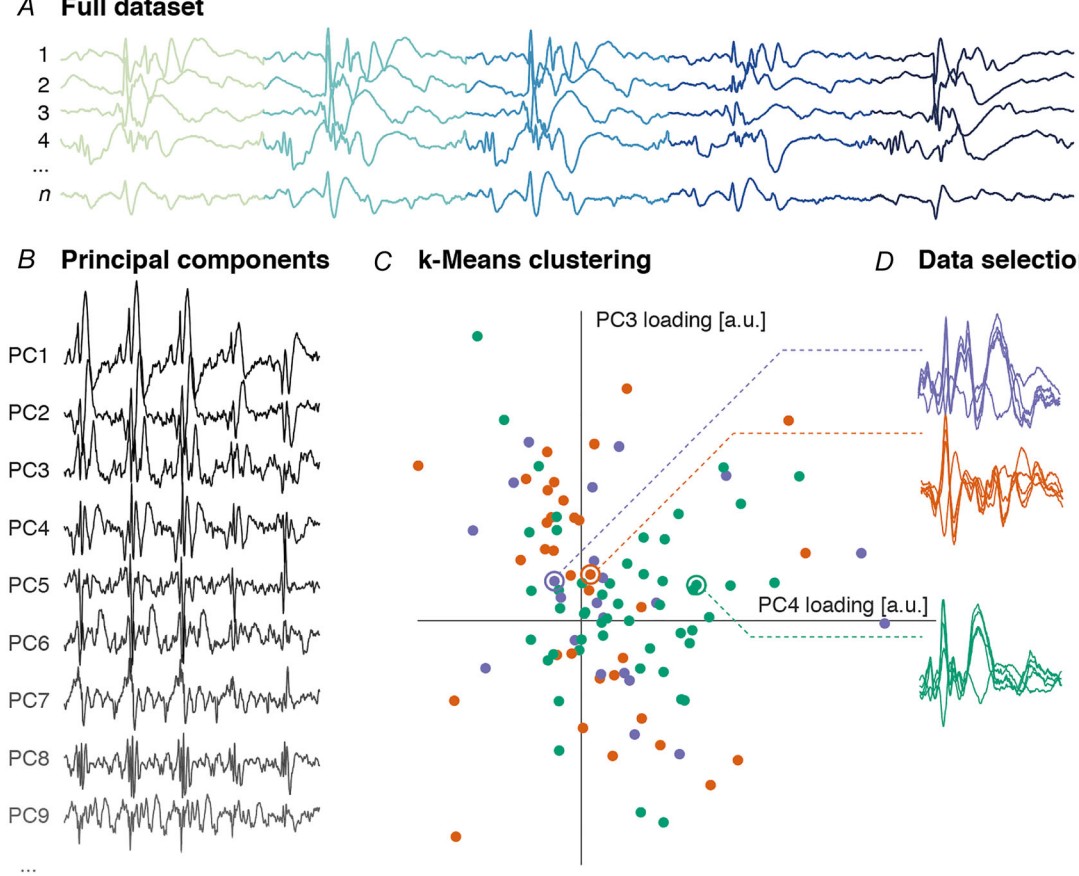

**Figure 2. Selection of representative interictal discharges for dynamic causal modelling**
*A*, full datasets were assembled into a single *n* by *t* matrix, where *n* is the total number of spike groups, and *t* is the number of temporal samples, concatenated across contacts. *B*, principal component analysis on this matrix was used to identify a subset of p components explaining >90% of the overall variance. Example components are shown, each of the length *t*. *C*, each individual spike group represents a point in a *p*-dimensional space, defined by the loadings on the principal components. We performed *k*-means clustering with a *k* = 3 in this space. *D* shows a two-dimensional projection of the low dimensional space in which IEDs were clustered. We then selected the data spike groups closest to each of the *k* centroids and retained them as representative examples of distributed IEDs for further dynamic causal modelling analysis. [Colour figure can be viewed at wileyonlinelibrary.com]

changes in intrinsic excitability, aiming to simulate possible transitions into ictal dynamics (Papadopoulou et al., 2015). Here, the DCM is used in 'simulation mode', meaning that, instead of inferring parameter values from given data, we simulated data based on fluctuations about empirically determined model parameters (i.e. synaptic connectivity and related time constants). We repeated these simulations with models that did and did not contain recurrent coupling between neighbouring nodes, aiming to illustrate the relationship between model topology and ensuing dynamics.

## Results

### Interictal discharges form travelling waves between tuber core and periphery

The distribution of spatially co-ordinated delays (Fig. 3) between spikes detected on multiple channels of the same SEEG electrodes were analysed for a total of >8000 individual spike groups across electrodes and subjects (median ~40 per electrode). These suggest that interictal discharges behave like a travelling wave.

We used the Bayesian information criterion, on a set of simple linear models, to identify the spatiotemporal dynamics of individual interictal discharges. The linear models characterise the relationship between relative delay and SEEG contact position as either (1) 'uniform' (i.e. no relationship between timing of spike detection and SEEG contact position); (2) 'depth to surface' (i.e. there is a gradient of delays between the deepest and the most superficial contact of the SEEG electrode); and (3) 'core to periphery' (i.e. there is a gradient of delays between contacts closest to the tuber core and contacts peripheral to this) (Fig. 3A). Models were estimated separately for low (1–13 Hz) and fast (13–120 Hz) frequency components to account for potential propagation differences between fast and slow components of the interictal epileptiform discharges.

The winning model at the group level (i.e. the one with the lowest BIC value) is the 'depth to surface' model for both frequency bands considered (Fig. 3A). The BIC difference is >800, greatly exceeding the commonly considered 'significance' threshold of $\Delta BIC > 10$ (Jones et al., 2001). This characterises IEDs as waves travelling between the tuber core and its periphery. This travel can also be identified on the level of individual spikes and on electrode-wide averages (Fig. 3B). There was no statistical relationship between propagation patterns at the individual SEEG electrodes and their clinically reported involvement in clinical seizure onset ($X^2$ (1, $N = 38$) = 1.59, $P = 0.207$).

### Interictal discharges are supported by recurrently coupled local networks

Using DCM, we compared how plausible local network architectures explain the observed spatiotemporal dynamics of the IEDs. We organised the model space along two main dimensions: (1) the origin of the IED, modelled here as input into one of three network nodes, and (2) two coupling schemes between neighbouring nodes in the network (Fig. 4A). For each SEEG electrode, a full model was fitted to a clustering-based selection of up to three IEDs per included tuber that maximally capture the diversity of IEDs observed in that electrode. The parameters and free energy of the full $3 \times 2 = 6$ model space were then estimated using Bayesian model reduction separately for each electrode, yielding independently parameterised sets of candidate models for each perituberal network. These models were then compared at the group-level (i.e. combining evidence across all patients) using a random effects analysis based on the model- and subject-specific free energy estimates of the model evidence. This provides an estimate of the group level exceedance probability that allows for comparison of explanatory models reported here (Fig. 4B).

Group level Bayesian parameter averages for the winning model are shown in Table 2. The winning model featured a triggering impulse at the tuber core, with recurrent forward/backward coupling to its neighbouring nodes. This further supports the observation that travelling wave dynamics organise around a core-periphery gradient, but adds the insight that recurrent coupling is required to support the observed IED spread. The winning model explained 58.0 ± 19.6% (mean ± SD) of the variance for each electrode, with large amplitude, lower frequency wave forms generally captured better (representative examples shown in Fig. 4C).

We further explored whether within-electrode differences in IEDs can be explained by changes in just a few key connections. For each electrode, DCMs were fitted with both a persistent set of coupling parameters, as well as modulations of this coupling for specific IEDs. To identify whether only changing a small subset of these parameter was sufficient to explain IED variability within tubers, we compared different constellations of this modulation; specifically comparing modulations of forward (*F*), backward (*B*) or intrinsic self-inhibitory connectivity (*i*) and their combinations. We did this again by performing Bayesian model reduction separately for each electrode, followed by group level random effects model comparison. However, to explain the diversity of IEDs observed patients, reciprocal extrinsic connections (*F* and *B*) and recurrent self-connections (*i*) were all required (Fig. 4D)

## A  Interictal discharge propagation in relation to tuber core

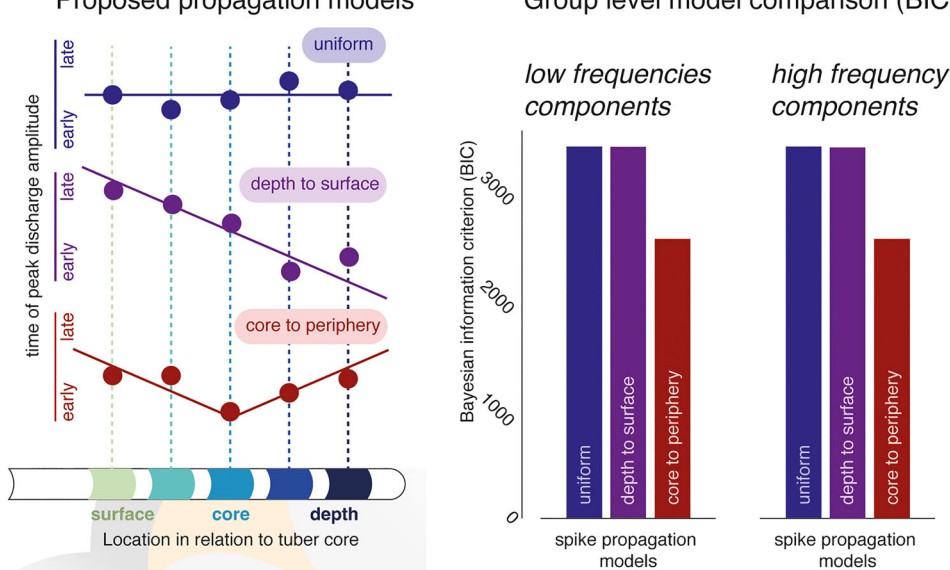

Proposed propagation models

Group level model comparison (BIC)

## B  Example data

Single interictal discharge example

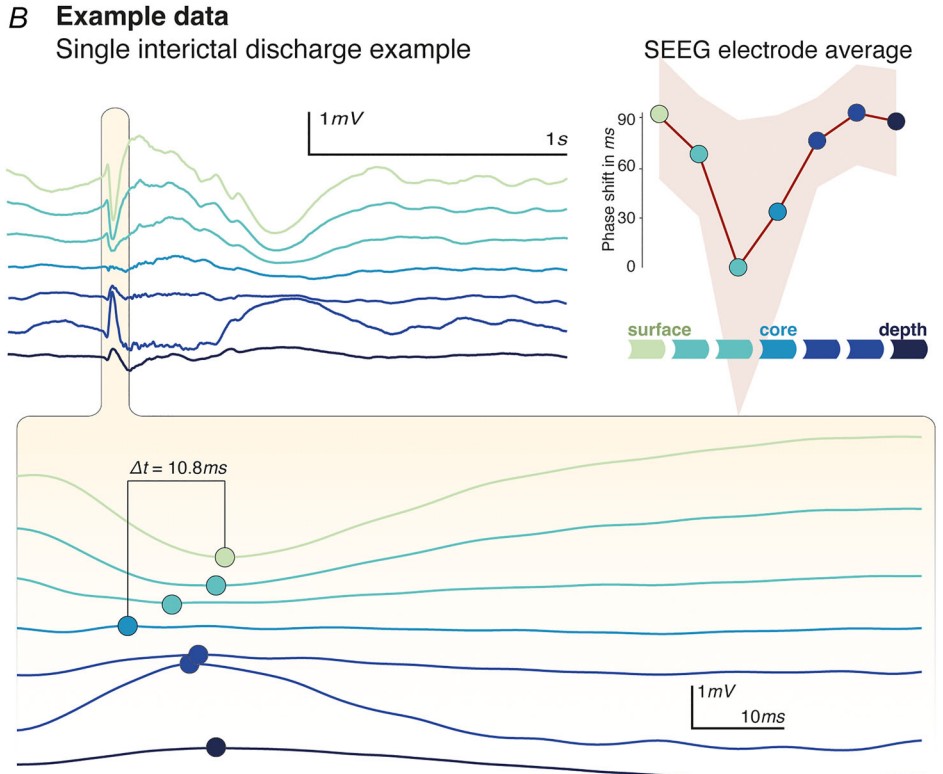

SEEG electrode average

**Figure 3. Delays follow a tuber core to periphery gradient**

*A*, three linear models were used to identify possible relationships between delays and spatial location of SEEG contacts. These explain the relationship either as uniform (no relationship), depth to surface or core to periphery local spread of putative IED travelling waves. Model comparison identifies the core to periphery organisation of IEDs as the most probable model (with the lowest BIC value) for both frequency components: low frequency BICs 3286.3, 3289.3 and 2461.0; high frequency BICs 3286.3, 3289.4 and 2462.4 for uniform, depth to surface and core to periphery models respectively. *B*, travelling wave dynamics can also be identified at the level of individual IEDs, as shown in the core to periphery organisation of temporal differences between spike peak timings (bottom). Note that because of the bipolar montage, not each of these peaks may be detected as a spike, and so the dots

indicate local maxima around the detected event. For the same electrode, the average temporal shift between detected groups of IEDs is shown for individual SEEG contacts against their spatial location in relation to the tuber core (shaded area illustrates standard deviation around the mean of 89 individual IEDs). [Colour figure can be viewed at wileyonlinelibrary.com]

### Only recurrently coupled network support both IED- and seizure-like dynamics in silico

The network models fitted in the previous step provide us with fully specified generative models, allowing for simulations under different parameter values. Here, we systematically varied the intrinsic excitability (i.e. gain) parameters of the neuronal populations (parameters $\gamma_{sp-sp}$, $\gamma_{sp-ss}$, $\gamma_{ii-ss}$) (Table 2) starting from a fully parameterised model from the previous step. Through the simulations, we identified a particular example DCM that supported fast, ictal-like dynamics in response to increasing gain mediated by those parameters. We then employed identical stepwise changes of intrinsic gain in the same network, but with the recurrent coupling switched off (i.e. only forward connectivity between neighbouring regions) (Fig. 5). These simulations reveal that, although fast frequency rhythms akin to ictal discharges may be produced by both recurrent and forward only network architecture, the IED-like dynamics are lost in the absence of the recurrent coupling, even in networks that still support the high frequency ictal-like discharges.

## Discussion

In the present study we have investigated the relationship of cortical tubers and surrounding perituberal cortex with interictal epileptiform discharges. We provide quantitative and modelling evidence that (1) tuber cores are the spatial source of interictal discharges; (2) IED travelling wave dynamics are most probably explained by locally recurrent tuber-perituberal networks; and (3) these networks can also generate hypersynchronous oscillations.

### Interictal discharges in tuberous sclerosis spread from the tuber core

This study confirms previous EEG findings that IEDs form travelling waves and propagate through the cerebral cortex (Alarcon et al., 1997; Emerson et al., 1995; Smith et al., 2022). IED delays are organised spatially in relation to the tuber core and IEDs travel at average speeds of 77–155 cm s$^{-1}$, broadly in line with propagations speeds previously reported (Smith et al., 2022). Modelling also identifies the interictal discharge impulse in the tuber core. Histologically, tuber cores show dysmorphic neurons and balloon cells, which are also found in other epileptogenic lesions such as focal cortical dysplasia type IIb (André et al., 2007). However, dyslamination and cellular dysplasia are also reported to affect the peri-tuberal cortex (Ruppe et al., 2014), providing a possible pathophysiological role for the peritubular cortex in ictogenesis. There has been conflicting evidence from SEEG and corollary investigations suggesting that seizures may either emerge from the core of the tuber (Kannan et al., 2016; Mohamed et al., 2012) or the surrounding cortex (Ma et al., 2012; Major et al., 2009).

We demonstrate that IEDs recapitulate a core-to-periphery organisation of abnormal brain dynamics, even outside of seizures. As in many cases of IEDs in patients with focal epilepsy, the relationship between IEDs and ictal dynamics at the individual level is non-trivial: We observed some variability, but the direction of local IED spread around individual tubers was not strongly associated with whether a tuber was involved at seizure onset.

### Recurrent local coupling is necessary for IED dynamics

As in previous work modelling IEDs (Wendling et al., 2024), we modelled with IEDs as perturbative transients in response to a spatially constrained external input. This approach does not capture the spontaneous, intermittent occurrence of discharges, but aims to capture the spatiotemporal distribution of discharges, once generated. Similar approaches have been useful in improving source localisation of epileptiform and physiological transients (Sun et al., 2022). Using a dynamic causal modelling approach, we aimed to (1) explore the onset and propagation of IEDs in relation to tuber anatomy and (2) identify the principles of network organisation that support the local spread of IEDs. We found evidence for a privileged role of the tuber core as origin of IEDs, which then spread along locally recurrent coupling between tuber core and perituberal cortex. This recurrent coupling confers coupled oscillatory dynamics on distributed responses, allowing for arbitrarily fast delays of travelling waves to emerge. This offers a novel perspective on the functional organisation of the local ictogenic network surrounding the tuber core. Although the tubers are apparently the origin of abnormal activity, it is their close, recurrent integration with surrounding cortex that allows for IED dynamics to emerge. A strongly recurrently coupled network such as this may itself support emergent dynamics such as seizures, even if a possible initial 'pacemaker' is eventually removed (e.g. through eventual calcification of the tuber core, or focal neurosurgical lesioning).

## *A*  Model space for IED propagation

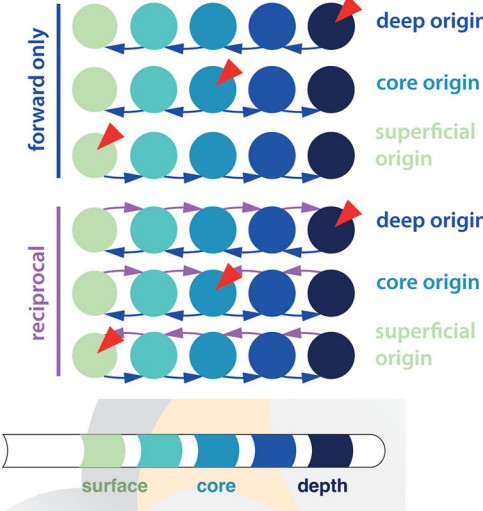

## *C*  Example spikes and model fits

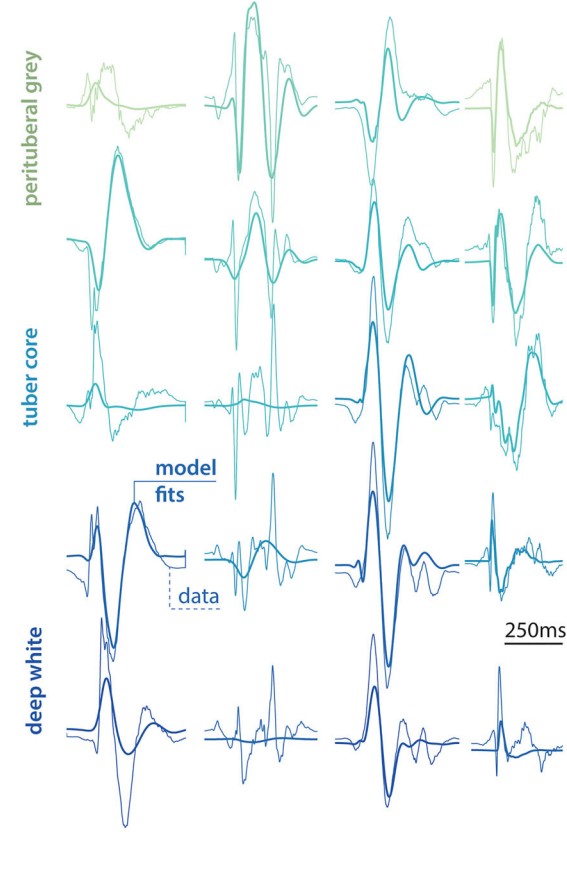

## *B*  Bayesian Model comparison

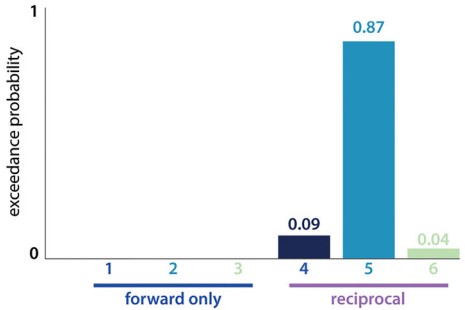

## *D*  Bayesian model comparison for differences in individual IEDs

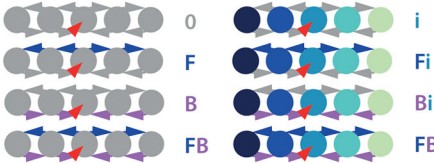
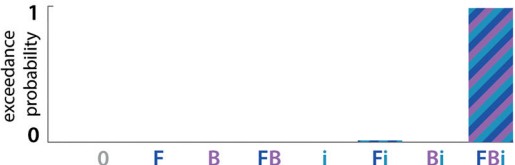

**Figure 4. Recurrent coupling in local networks supports IED**
*A*, six candidate networks were fitted separately to representative IEDs recorded around each tuber: three sets of forward coupled, or recurrently coupled networks, each with either superficial, core or deep origin of the IED. *B*, Bayesian model comparison shows that recurrent coupling is necessary to explain IED dynamics and that across the group spread from the tuber core was the most probable local network architecture. *C*, the final winning model captured an average of ∼60% of the variance of single spike dynamics across patients, with some representative model fits shown here. *D*, additionally, we explored which kind of within-electrode modulations of coupling are required to explain the variability between the representative IEDs modelled. For this, we used the winning (recurrent) model architecture from the previous step, in which between IED changes in all coupling parameters were allowed. We then restricted the types of parameter changes that could explain between IED differences within a single electrode to forward coupling (F), backward coupling (B) or intrinsic self-inhibitory coupling (1) changes, or a combination of those; all models for within-electrode between-IED variation are shown on the left. Group level random effects Bayesian model comparison is shown on the right, indicating that changes in all parameters are required to explain the observed variety of IEDs within electrodes. [Colour figure can be viewed at wileyonlinelibrary.com]

## Coupled-oscillator dynamics support both ictal- and IED dynamics

Simulations based on empirically fitted dynamic causal models allowed us to reproduce (*in silico*) a transition from IED dynamics to fast oscillation onset rhythm, akin to low voltage hypersynchronous fast activity in focal seizure onset. These simulations demonstrate that only the networks with locally recurrent coupling support both IED-like discharges, as well as the high frequency ictal-like rhythms. This simulation illustrates the fact that two dynamical regimes can emerge in a single network: the ictal wave front is often characterised by fast frequency rhythms that slowly recruit more cortical areas over time, whereas later ictal spikes in an established seizure can propagate at near synchronous speeds in the areas of cortex that have already been recruited (Schevon et al., 2019; Smith et al., 2016). These different types of propagation mechanisms can be described in different models, such as propagation in excitable media (ictal onset), or coupled oscillator dynamics (for ictal spike and wave discharges).

Recent computational modelling work has shown that variations in slowly evolving states within a single heterogeneously coupled network of neural masses can support both types of dynamics by transitioning through different types of dynamical regimes during a modelled seizure (Proix et al., 2018). The work presented here suggests that the network architecture that supports these diverse dynamics may already be in place during IEDs.

### Limitations

There are several important limitations to the present study. We are fundamentally constrained by the data that are available; although SEEG allows a unique insight into *in vivo* dynamics of the epileptic brain, the recordings are indirect, macroscopic summaries of a large number of individual neurons. Inferring synaptic parameters at this macroscale may therefore be limited. Dynamic causal modelling offers one approach to address this ill-posed problem (i.e. inferring synaptic parameters from summary statistics of neuronal activity, such as the EEG signal). However, because of computational constraints, this inference relies on expectation-maximisation and local gradient descent, meaning that it does not always identify a globally optimal parameter, given a dataset, but may get stuck in local minima. Second, the modelling is dependent on choice of network structures included in the model comparison, and the choice of prior values. Future updates on, for example, optimal priors for parameters for intracranial EEG recording may therefore alter some of

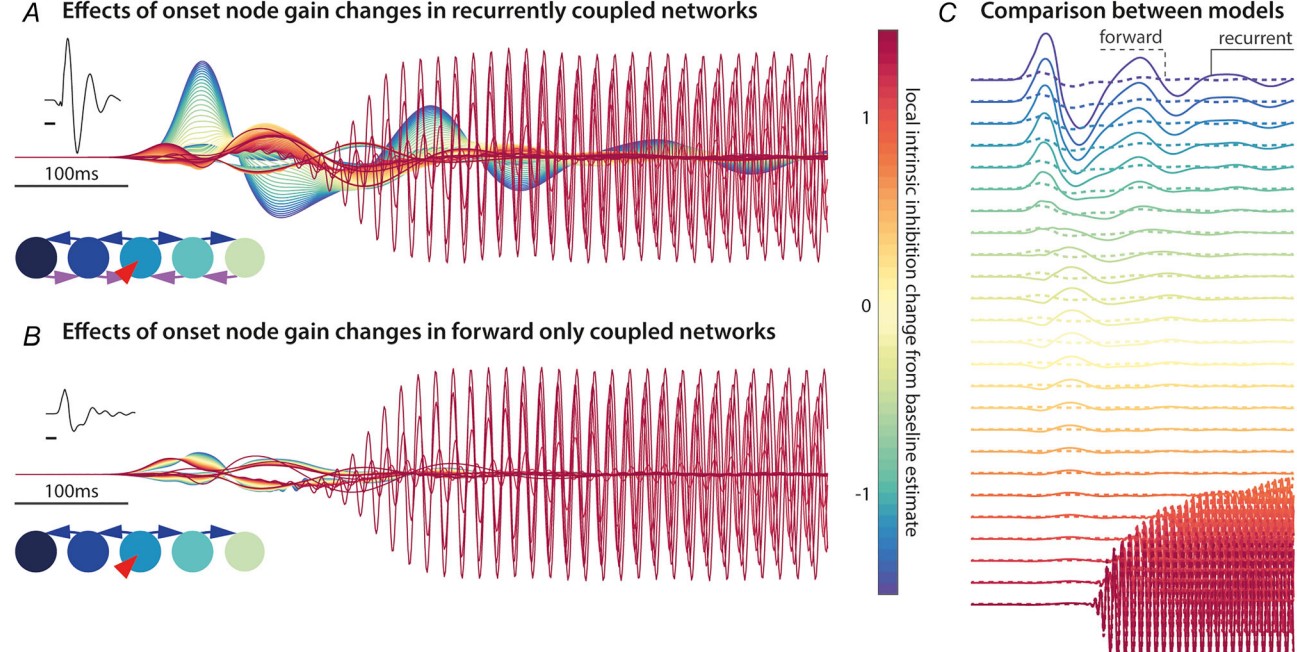

**Figure 5. Only networks with recurrent coupling support IED- and seizure-like dynamics**
*A*, using a single IED-generating network model fitted to one electrode dataset as template, we simulate changes in local intrinsic gain of the IED onset node (here the tuber core). Increases in gain allow for high frequency oscillations in the gamma range (40–50 Hz) akin to seizure-onset rhythms to emerge. *B*, using a model without recurrent coupling reproduces the fast frequency ictal-like discharges, but not the high amplitude synchronous IED-like discharges. *C*, the direct comparison between these simulated time series shows the large amplitude difference at low intrinsic gain values. [Colour figure can be viewed at wileyonlinelibrary.com]

the inference presented here. Additionally, SEEG contacts are placed according to clinical need and neurosurgical access, and they do not comprehensively map the entire cortex and do not have identical trajectories across tubers. In the present study, this is addressed by visual labelling of individual SEEG contacts in reference to the respective tuber that is being targeted. However, even with this labelling, the distances used for are the direct, Euclidean distance between individual contacts, whereas neuronal connections may have to traverse a more circuitous route. We may therefore underestimate the effective distances between SEEG contacts in our analysis here. In the future, including, for example, patient-specific structural connectivity data may provide neurobiologically more informed models that can be tested against each other in the modelling step (Sokolov et al., 2018). Furthermore, individual tubers in this analysis are treated independently of one another, even though, within one patient, many of these tubers are connected through longer distance, albeit less dense, connections.

Although these limitations currently affect how well this approach will translate to help localise sources of epileptogenic activity in individual patients, the principal recurrent architecture with tuber core origin of epileptiform activity could be demonstrated clearly as a superior macroscopic model of tuber–perituberal network architecture.

## Implications

Our study used TSC as an interesting test case to explore the local network constraints of interictal dynamics because patients share an underlying aetiology and often have multiple lesions that can be targeted during the same SEEG recording. Also, there are important clinical questions in relation to the involvement of perituberal cortex in ictogenesis that remain to be addressed. Patients with tuberous sclerosis have constant growth of new tubers and maturation (calcification) of already existing tubers (Curatolo et al., 2008; Grajkowska et al., 2010), meaning that not previously ictogenic tubers can become ictogenic over time, and that new tubers can form, leading to new epileptic foci. Thus, improving our surgical approaches by carefully restricting surgical resections to the most probable epileptogenic regions and thereby limiting excess morbidity is essential because patients with TSC may undergo several epilepsy surgeries during their lifetime (Bollo et al., 2008; Fallah et al., 2015). A detailed understanding of interictal and ultimately seizure dynamics in tubers, perituberal cortex and unaffected cortex is an essential step towards improving the surgical approaches. Future work will also need to integrate the network effects of multiple tubers in the same patient and their interaction as an epileptogenic network, which is an extension of the present work beyond the local networks around individual tubers.

Although our results specifically address patients with TSC, other pharmaco-resistant epilepsies are associated with pathologies in the same molecular (mechanistic target of rapamycin) pathway (e.g. focal cortical dysplasia type 2) and share histological features; therefore, our results may also be applicable to that group of patients (Liu et al., 2014; Marsan & Baulac, 2018; Meng et al., 2013).

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

## Additional information

### Data availability statement

Where possible without compromising the anonymity of the patients, data are available upon reasonable request. This may include, for example, epoched recordings of interictal epileptiform discharge groups with contacts labelled in reference to the lesion. All custom code is available freely online (https://github.com/roschkoenig/travelling_spikes).

## Competing interests

The authors declare that they have no competing interests.

## Author contributions

S.T and R.E.R were responsible for formal analysis. S.T, R.T, M.T and R.E.R were responsible for investigations. S.T, R.T and M.T were responsible for data curation. S.T and R.E.R were responsible for writing the original draft. R.T, M.T, T.B, K.J.F and R.E.R were responsible for conceptualisation. R.T, M.T, T.B, K.J.F and R.E.R were responsible for reviewing and editing. R.T, M.T and R.E.R were responsible for funding acquisition. K.J.F and R.E.R were responsible for methodology. R.E.R was responsible for visualisation and supervision.

## Funding

This work is supported by the National Institute for Health Research Biomedical Research Centre at Great Ormond Street Hospital for Children NHS Foundation Trust and University College London. The project was funded by the Oakgrove Charitable Foundation (MT, RT, ST, RER), and the Wellcome Trust (RER: 209 164/Z/17/Z, 106 556/Z/14/Z, KJF: 08 8130/Z/09/Z)

## Acknowledgements

We are grateful to all the patients and their families, as well as to the clinical teams overlooking their care.

## Keywords

computational modelling, dynamic causal modelling, epilepsy, epileptogenic networks, intracranial EEG, interictal epileptiform activity, tuberous sclerosis

## Supporting information

Additional supporting information can be found online in the Supporting Information section at the end of the HTML view of the article. Supporting information files available:

**Peer Review History**

