## [Peer Review History · The Journal of Physiology]

Interictal discharges spread along local recurrent networks between tubers and surrounding cortex

Stasa Tumpa, Rachel Thornton, Martin Tisdall, Torsten Baldeweg, Karl Friston, and Richard Ewald Rosch
DOI: 10.1113/JP288141

Corresponding author(s): Richard Rosch (richard.rosch@kcl.ac.uk)

The following individual(s) involved in review of this submission have agreed to reveal their identity: Rikkert Hindriks (Referee #1); Huyfang Wang (Referee #2); Ken D O'Halloran (Referee #3)

Review Timeline:

Submission Date:	19-Nov-2024
Editorial Decision:	07-Jan-2025
Revision Received:	24-Jan-2025
Editorial Decision:	17-Feb-2025
Revision Received:	18-Feb-2025
Accepted:	21-Feb-2025

Senior Editor: Richard Carson

Reviewing Editor: Bettina Schwab

Transaction Report:

Dear Dr Rosch,

Re: JP-RP-2024-288141 "Interictal discharges spread along local recurrent networks between tubers and surrounding cortex" by Stasa Tumpa, Rachel Thornton, Martin Tisdall, Torsten Baldeweg, Karl Friston, and Richard Ewald Rosch

Thank you for submitting your manuscript to The Journal of Physiology. It has been assessed by a Reviewing Editor and by 2 expert referees and we are pleased to tell you that it is potentially acceptable for publication following satisfactory major revision.

REVISION CHECKLIST:

Please upload two versions of your manuscript text: one with all relevant changes highlighted and one clean version with no

changes tracked. The manuscript file should include all tables and figure legends, but each figure/graph should be uploaded as separate, high-resolution files.

We look forward to receiving your revised submission.

Yours sincerely,

Richard Carson
Senior Editor
The Journal of Physiology

REQUIRED ITEMS

- Include a Key Points list in the article itself, before the Abstract.
- Author photo and profile. First or joint first authors are asked to provide a short biography (no more than 100 words for one author or 150 words in total for joint first authors) and a portrait photograph. These should be uploaded and clearly labelled together in a Word document with the revised version of the manuscript. See Information for Authors for further details.
- You must start the Methods section with a paragraph headed Ethical Approval. If experiments were conducted on humans, confirmation that informed consent was obtained, preferably in writing, that the studies conformed to the standards set by the latest revision of the Declaration of Helsinki and that the procedures were approved by a properly constituted ethics committee, which should be named, must be included in the article file. If the research study was registered (clause 35 of the Declaration of Helsinki), the registration database should be indicated, otherwise the lack of registration should be noted as an exception (e.g. The study conformed to the standards set by the Declaration of Helsinki, except for registration in a database). For further information see: <https://physoc.onlinelibrary.wiley.com/hub/human-experiments>.
- The reference list must be in alphabetical order, rather than numbered, to comply with our Journal format.
- Your manuscript must include a complete Additional Information section, including competing interests; funding; author contributions and acknowledgements.
- Please upload separate high-quality figure files via the submission form.
- Your paper contains Supporting Information of a type that we no longer publish, including supplementary tables and figures. Any information essential to an understanding of the paper must be included as part of the main manuscript and figures. The only Supporting Information that we publish are video and audio, 3D structures, program codes and large data files. Your revised paper will be returned to you if it does not adhere to our Supporting Information Guidelines.
- We invite you to include a Translational Perspective paragraph in your manuscript. This should be included in the main body of the manuscript after the Acknowledgements. It should describe the wider translational implications of the work, in plain English, for a broad scientific audience. Please use the following guidelines to prepare a Translational Perspective of

your paper: https://jp.msubmit.net/cgi-bin/main.plex?form_type=display_requirements#authortranspersp. The Translational Perspective should not exceed 250 words in total and should be presented as a single paragraph. Abbreviations and technical terms must be defined as briefly and simply as possible the first time they are used, unless they are generally/easily understood, e.g. ECG, HIV/AIDS, K+ channel. Use language that can be understood by scientists or clinicians with a general knowledge of the topic addressed. Ensure the paragraph includes the hypothesis tested in the paper and accurately reflects the findings of the paper and the implications for future research. Please state the word count of the Translational Perspective paragraph.

- A Data Availability Statement is required for all papers reporting original data. This must be in the Additional Information section of the manuscript itself. It must have the paragraph heading 'Data Availability Statement'. All data supporting the results in the paper must be either: in the paper itself; uploaded as Supporting Information for Online Publication; or archived in an appropriate public repository. The statement needs to describe the availability or the absence of shared data. Authors must include in their statement: a link to the repository they have used, or a statement that it is available as Supporting Information; reference the data in the appropriate sections(s) of their manuscript; and cite the data they have shared in the References section. Whenever possible, the scripts and other artefacts used to generate the analyses presented in the paper should also be publicly archived. If sharing data compromises ethical standards or legal requirements then authors are not expected to share it, but must note this in their statement. For more information, see our Statistics Policy.

- Please include an Abstract Figure file, as well as the Figure Legend text within the main article file. The Abstract Figure is a piece of artwork designed to give readers an immediate understanding of the research and should summarise the main conclusions. If possible, the image should be easily 'readable' from left to right or top to bottom. It should show the physiological relevance of the manuscript so readers can assess the importance and content of its findings. Abstract Figures should not merely recapitulate other figures in the manuscript. Please try to keep the diagram as simple as possible and without superfluous information that may distract from the main conclusion(s). Abstract Figures must be provided by authors no later than the revised manuscript stage and should be uploaded as a separate file during online submission labelled as File Type 'Abstract Figure'. Please also ensure that you include the figure legend in the main article file. All Abstract Figures should be created using BioRender. Authors should use The Journal's premium BioRender account to export high-resolution images. Details on how to use and access the premium account are included as part of this email.

EDITOR COMMENTS

Reviewing Editor:

Thank you for submitting your work to The Journal of Physiology. Your manuscript was reviewed by two experts, who both highlight the quality of your work but also have some concerns that need to be addressed. In particular, please do not refer to supplementary items, but include everything that is relevant in the main manuscript. Furthermore, as also mentioned already by reviewer 1, the limitation section should be expanded and include limitations of the dynamic causal modeling approach.

Please also see 'Required Items' above.

REFEREE COMMENTS

Referee #1:

In the submitted manuscript, the authors analyze and model intracranial EEG data from tuberous sclerosis patients, to better understand the dynamical mechanisms underlying propagation of interictal epileptiform discharges (IEDs). They focused on two aspects: location of origin and type of coupling (feed-forward or recurrent). Both the analysis and modeling supported the conclusion that IEDs start in the tuber and propagate via recurrent connections.

Both the analysis and modeling are carefully done, in particular with respect to the statistical aspects, and I find the main conclusion convincing. I don't have serious methodological reservations.

The manuscript is very well written, has a clear overall structure, clear descriptions of the results, the introduction provides

sufficient background, and discussion provides sufficient context (but see minor comment 9 and 10) and the figures and illustrations are of high quality and very illustrative.

MAJOR COMMENTS

1. The supplementary material was not uploaded to the submission system. Since the journal doesn't allow supplementary material, perhaps the authors can include it in the main text (in shortened form).

2. The models are fitted to up to three IEDs per included tuber (line 278). However, more than 8000 groups of IEDs were identified in the data (line 254). So it's not clear to me why only a very small subset of these was used for fitting. The concern is that the fitting results might not be representative for the entire data.

MINOR COMMENTS

1. The description of the delay estimation (Section 2.3.1) is incomplete: How exactly is the delay estimated from the cross-correlation function? Is this done by taking the lag at which the cross-correlation function has a local maximum?

2. The authors refer to the delay as a "phase-delay", but I believe this is only appropriate if it is measured in radians, whereas the delays in this study are measured in time units (ms).

3. The description of the linear models (Section 2.3.2) is incomplete. In particular, the statistical model for the observations is not specified. Without knowing the model, the likelihood function cannot be determined. It will be good to include the probability distribution of the observations and the likelihood functions.

This allows the reader to calculate the BIC values.

4. Perhaps the authors can briefly comment on the observed propagation speeds of the low-frequency oscillations and IEDs and whether these are in line with earlier studies.

5. In the analysis reported in the last paragraph of Section 3.2, matrices A and B are introduced, but not explained. Are they perhaps described in the supplementary material?

6. Line 205: It is not entirely clear what the authors mean by "under Bayesian constraints". It seems that this phrase can be omitted.

7. Line 290: The authors write that, on average, the winning model explained 58% of the variance. I would be interested to know how variable this percentage is over electrodes. So perhaps the authors could add the standard deviation over electrodes?

8. To be sure: Are the fitted models situated in a fixed-point regime? And is this in line with previous modeling studies on IEDs? Perhaps the authors can briefly discuss this. This is merely a suggestion.

9. Section 4.4 (Limitations) is not worked out well. I would suggest to either delete it or work it out in more detail.

10. Section 4.5 (Implications) It doesn't become entirely clear how the current study contributes to improved surgical approaches. Perhaps the authors can be a bit more concrete here?

Sections 4.4 and 4.5 give the impression that they are hastily written.

GRAMMATICAL

Line 67: "canx" -> "can"

Line 165: "of a spike-wave discharges" -> "of a wave-spike-wave discharge"

Line 174: "could explain spatiotemporal" -> "could explain the spatiotemporal"

Line 176: A closing bracket is missing.

Line 205: A period is missing and "Where" -> "When"

Line 214: "use the dynamic" -> "use dynamic"

The authors use both "SEEG" and "iEEG"

Line 232: "most superficial superficial" -> "most superficial layer"?

Legend to figure 4: "fitted to one electrode dataset fitted to one electrode dataset"

Line 207: "network support" -> "networks support"

Line 326: "network" -> "networks"

Line 386: "from during".

Please check the entire text for errors.

Referee #2:

This paper investigated the network mechanisms that underlie the generation and spread of interictal epileptiform discharges (IEDs) using neural mass modeling of cortical dynamics based on SEEG recordings in epileptic patients with tuberous sclerosis.

I enjoyed reading this paper. It is well-structured, well-written, and easy to follow. Below are some comments that I hope will help the authors improve their paper:

1. The authors mentioned that there are separate and multiple tubers within an individual patient. What is the relationship among these tubers? The tuber cores are important for the analysis in this paper. Does each tuber have a core that is analyzed? In Section 4.1, the authors discussed the relationship between IEDs and ictal dynamics, as well as IEDs and seizure onsets. It seems that all tuber cores act as sources within their local networks. Do they exhibit any network

connectivity or relationships among them? Is this an important question to discuss further?

2. In Abstract: Please add at least one sentence to clarify the main results, such as: "Tuber cores are the spatial sources of interictal discharges, and IED traveling wave dynamics are most likely explained by locally recurrent tuber-perituberal networks."

3. I could not figure out the implications of these results for understanding the network mechanisms that might inform surgical decisions or approaches. Perhaps the authors could elaborate more on this point in Section 4.5.

4. Table 1: Please Consider adding an additional column to indicate the number of tubers for each patient.

5. Figure 1: There is an impulse shown in the figure, which should be mentioned in the caption as well.

6. In Equation 1: in my opinion, the correct description of the relationship between t and p should be: the variable time t is independent of position p . I don't think that b_{uniform} defines a constant relationship between position and time as the authors claimed. A linear relationship can be constant, but that is not the case here.

END OF COMMENTS

We are grateful for the reviewers encouraging and insightful comments and hope that the changes made to the manuscript address all the comments outlined below. We have included a point-by-point response to reviewers, with changes made to the manuscript highlighted in red.

Referee #1:

In the submitted manuscript, the authors analyze and model intracranial EEG data from tuberous sclerosis patients, to better understand the dynamical mechanisms underlying propagation of interictal epileptiform discharges (IEDs). They focused on two aspects: location of origin and type of coupling (feed-forward or recurrent). Both the analysis and modeling supported the conclusion that IEDs start in the tuber and propagate via recurrent connections.

Both the analysis and modeling are carefully done, in particular with respect to the statistical aspects, and I find the main conclusion convincing. I don't have serious methodological reservations.

The manuscript is very well written, has a clear overall structure, clear descriptions of the results, the introduction provides sufficient background, and discussion provides sufficient context (but see minor comment 9 and 10) and the figures and illustrations are of high quality and very illustrative.

We really appreciate the time taken to review our paper and this encouraging summary.

MAJOR COMMENTS

1. The supplementary material was not uploaded to the submission system. Since the journal doesn't allow supplementary material, perhaps the authors can include it in the main text (in shortened form).

Supplementary material has now been integrated into the main paper - thank you for highlighting this oversight. This includes additions to the Methods section in '2.4.1 DCM framework' and a new section '2.4.2 Bayesian model reduction and model comparison'

2. The models are fitted to up to three IEDs per included tuber (line 278). However, more than 8000 groups of IEDs were identified in the data (line 254). So it's not clear to me why only a very small subset of these was used for fitting. The concern is that the fitting results might not be representative for the entire data.

Thank you for highlighting this - the now included (previously supplementary) material should now address this. We have carefully selected 3 'representative' IEDs from the median ~40 IEDs per electrode using a clustering approach to identify representative samples from diverse (in morphology and channel distribution) groups of IEDs. This approach (rather than e.g. averaging) was chosen in order not to 'blunt' features of the IEDs. We have included the following relevant section in the Methods:

In standard ERP applications of DCM, in order to improve signal to noise ratio, averages of many instances of a cortical potential are used as the data features. In our case, we found that averaging IEDs within an electrode could blur the shape of the discharges, due to small variations in their morphology across the sample. We therefore opted to model individual IEDs (rather than averages). In a trade-off between computational load of the analysis, and capturing the maximum variability present in the data set, we used a data-driven clustering approach that helped us select a set of three IEDs per patient that were representative of the whole dataset as follows (this can be regarded as a form of robust averaging within similar response classes): (1) We performed a principal component analysis of the multichannel recordings of all individual IEDs in a single SEEG electrode and retained the components necessary to explain 90% of the variance in the dataset. (2) In this low dimensional representation of the data, we performed k-means clustering, to identify groups of IEDs that were morphologically similar, whilst being as distinct as possible from other groups. (3) Having identified the centroids of these clusters, we then selected the IED closest (in principal component space) to the centroid as representative of the group for subsequent inversion. See Figure 2 for a worked example. This resulted in three representative IEDs for each tuber.

Figure 2 – Selection of representative interictal discharges for dynamic causal modelling. A) Full datasets were assembled into a single n by t matrix, where n is the total number of spike groups, and t is the number of temporal samples, concatenated across contacts. (B) Principal component analysis on this matrix was used to identify a subset of p components explaining $>90\%$ of the overall variance. The figure shows example components, each of the length t . (C) Each individual spike group represents a point in a p -dimensional space, defined by the loadings on the principal components. We performed k -means clustering with a $k = 3$ in this space (of which a two-dimensional projection is shown here). We then selected the data spike groups closest to each of the k centroids, and retained them as representative examples of distributed IEDs for further dynamic causal modelling analysis.

MINOR COMMENTS

1. The description of the delay estimation (Section 2.3.1) is incomplete: How exactly is the delay estimated from the cross-correlation function? Is this done by taking the lag at which the cross-correlation function has a local maximum?

Correct. We have now updated this as follows:

We estimated the temporal delay between clusters of spikes spanning several channels of the same SEEG electrode, by calculating the cross-correlation between the signal, and estimating the delay between neighbouring channels **at which cross-correlation is maximal**

2. The authors refer to the delay as a "phase-delay", but I believe this is only appropriate if it is measured in radians, whereas the delays in this study are measured in time units (ms).

We have updated the text to refer to 'delay' rather than 'phase delay'.

3. The description of the linear models (Section 2.3.2) is incomplete. In particular, the statistical model for the observations is not specified. Without knowing the model, the likelihood function cannot be determined. It will be good to include the probability distribution of the observations and the likelihood functions. This allows the reader to calculate the BIC values.

All models and BIC calculation are fully specified in the code that is available online - we have now referenced this more clearly in the relevant methods section:

The model comparison is implemented in Matlab with custom scripts accessible (https://github.com/roschkoenig/Travelling_Spikes) and specification of models contained in the `ts_delaystats.m` script.

4. Perhaps the authors can briefly comment on the observed propagation speeds of the low-frequency oscillations and IEDs and whether these are in line with earlier studies.

Our findings are in line with those previously reported, which we have now included in the discussion.

Discussion

IED delays are organised spatially in relation to the tuber core and IEDs travel at average speeds of 77-155cm/s, broadly in line with propagation speeds previously reported (56).

5. In the analysis reported in the last paragraph of Section 3.2, matrices A and B are introduced, but not explained. Are they perhaps described in the supplementary material?

Yes these are included in the supplementary methods - but actually bear no importance here, so for ease of understanding, we have now removed reference to the matrices.

6. Line 205: It is not entirely clear what the authors mean by "under Bayesian constraints". It seems that this phrase can be omitted.

Agreed - we have now removed this

7. Line 290: The authors write that, on average, the winning model explained 58% of the variance. I would be interested to know how variable this percentage is over electrodes. So perhaps the authors could add the standard deviation over electrodes?

This has now been included as follows in the results

The winning model explained an average of 58.0% (standard deviation 19.6%) of the variance for each electrode

8. To be sure: Are the fitted models situated in a fixed-point regime? And is this in line with previous modeling studies on IEDs? Perhaps the authors can briefly discuss this. This is merely a suggestion.

Correct, we have now included this in the Methods, and discussed a number of relevant papers in the Discussion.

In the Methods

We perform this analysis in a fixed-point regime, where IEDs are modeled as transient, provoked perturbations that elicit damped oscillatory dynamics around the stable fixed point attractor of the specified model.

In the Discussion

As in previous work modelling IEDs (59), we modelled with IEDs as perturbative transients in response to a spatially constrained external input. This approach does not capture the spontaneous, intermittent occurrence of discharges, but aims to capture the spatiotemporal distribution of discharges, once generated. Similar approaches have been useful in improving source localisation of epileptiform, and physiological transients (60)

9. Section 4.4 (Limitations) is not worked out well. I would suggest to either delete it or work it out in more detail.

We have now updated Limitations as follows:

There are several important limitations to the study. We are fundamentally constrained by the data that is available – although SEEG allows a unique insight into in vivo dynamics of the epileptic brain, the recordings are indirect, macroscopic summaries of a large number of individual neurons. Inferring synaptic parameters at this macroscale may therefore be limited. Additionally, SEEG contacts are placed according to clinical need and neurosurgical access, and do not comprehensively map the entire cortex, and do not have identical trajectories across tubers. In the current study this is addressed by visual labelling of individual SEEG contacts in reference to the respective tuber that is being targeted. However even with this labelling, the distances used for the current analysis are the direct, Euclidean distance between individual contacts, whilst neuronal connections may have to traverse a more circuitous route. We may therefore underestimate the effective distances between SEEG contacts in our analysis here. In future, including, for example, patient-specific structural connectivity data may provide neurobiologically more informed models that can be tested against each other in the modelling step (62). Furthermore, individual tubers in this analysis are treated independently of one another, even though within one patient, many of these tubers are connected through – albeit less dense – longer distance connections. Whilst these limitations currently affect how well this approach will translate to help localise sources of epileptogenic activity in individual patients, the principal recurrent architecture with tuber core origin of epileptiform activity could be demonstrated clearly as a superior macroscopic model of tuber - perituberal network architecture.

10. Section 4.5 (Implications) It doesn't become entirely clear how the current study contributes to improved surgical approaches. Perhaps the authors can be a bit more concrete here?

We have now specified this further

Thus improving our surgical approaches by carefully restricting surgical resections to the most likely epileptogenic regions thereby limiting excess morbidity is essential, as patients with TSC may undergo several epilepsy surgeries during their lifetime (13,14). A detailed understanding of interictal, and ultimately seizure dynamics in tubers, perituberal cortex, and unaffected cortex is an essential step towards improving the surgical approaches.

Sections 4.4 and 4.5 give the impression that they are hastily written.

We hope that the updates to these sections have improved them and made them more relevant.

GRAMMATICAL

Line 67: "canx" -> "can" corrected

Line 165: "of a spike-wave discharges" -> "of a wave-spike-wave discharge" - we have now adapted this to the ACNS terminology of 'spike and wave discharge' and put it into singular

Line 174: "could explain spatiotemporal" -> "could explain the spatiotemporal" corrected

Line 176: A closing bracket is missing. corrected

Line 205: A period is missing and "Where" -> "When" corrected

Line 214: "use the dynamic" -> "use dynamic" corrected

The authors use both "SEEG" and "iEEG" corrected

Line 232: "most superficial superficial" -> "most superficial layer"? corrected

Legend to figure 4: "fitted to one electrode dataset fitted to one electrode dataset" corrected

Line 207: "network support" -> "networks support" corrected

Line 326: "network" -> "networks" corrected

Line 386: "from during". corrected

Please check the entire text for errors.

Referee #2:

This paper investigated the network mechanisms that underlie the generation and spread of interictal epileptiform discharges (IEDs) using neural mass modeling of cortical dynamics based on SEEG recordings in epileptic patients with tuberous sclerosis.

I enjoyed reading this paper. It is well-structured, well-written, and easy to follow. Below are some comments that I hope will help the authors improve their paper:

Thank you for taking the time for providing this review and the encouraging remarks. Very much appreciated.

1. The authors mentioned that there are separate and multiple tubers within an individual patient. What is the relationship among these tubers? The tuber cores are important for the analysis in this paper. Does each tuber have a core that is analyzed? In Section 4.1, the authors discussed the relationship between IEDs and ictal dynamics, as well as IEDs and seizure onsets. It seems that all tuber cores act as sources within their local networks. Do they exhibit any network connectivity or relationships among them? Is this an important question to discuss further?

This becomes a really essential question when thinking about the implication of these findings for epilepsy surgery - which has taken into account not just individual tubers, but also the relationship between them. This was out of scope of the current study, but we have added the following the the 'Limitations' and 'Implications'

In the Limitations

Furthermore, individual tubers in this analysis are treated independently of one another, even though within one patient, many of these tubers are connected through – albeit less dense – longer distance connections.

In the Implications

Future work will also need to integrate the network effects of multiple tubers in the same patient and their interaction as an epileptogenic network, which is an extension of the present work beyond the local networks around individual tubers.

2. In Abstract: Please add at least one sentence to clarify the main results, such as: "Tuber cores are the spatial sources of interictal discharges, and IED traveling wave dynamics are most likely explained by locally recurrent tuber-perituberal networks."

We have made the following changes to the abstract

Our results indicate that tuber cores are the spatial sources of interictal discharges, and interictal epileptiform discharge traveling wave dynamics are most likely explained by locally recurrent tuber-perituberal networks. This view integrates competing theories regarding the pathological organisation of epileptic foci and surrounding cortex in patients with tuberous sclerosis

3. I could not figure out the implications of these results for understanding the network mechanisms that might inform surgical decisions or approaches. Perhaps the authors could elaborate more on this point in Section 4.5.

This has now been adapted as follows

Thus improving our surgical approaches by carefully restricting surgical resections to the most likely epileptogenic regions thereby limiting excess morbidity is essential, as patients with TSC may undergo several epilepsy surgeries during their lifetime (13,14). A detailed understanding of interictal, and ultimately seizure dynamics in tubers, perituberal cortex, and unaffected cortex is an essential step towards improving the surgical approaches.

4. Table 1: Please Consider adding an additional column to indicate the number of tubers for each patient.

This is not easily done, because most patients have innumerable tubers, not all of which will be captured in detail, even in the radiology reports. For the purposes of SEEG, tubers are implanted that are likely a surgical target - usually based on multidisciplinary clinical planning depending on seizure semiology, scalp EEG findings during previous seizures, and radiological appearances of the tuber. Therefore even the number of tubers implanted per patient does not necessarily reflect a single quality.

5. Figure 1: There is an impulse shown in the figure, which should be mentioned in the caption as well.

We have adapted the caption as follows:

(B) Dynamic causal modelling allows the testing of network models of dynamics that generate SEEG responses to endogenous fluctuations or spikes, in this case modelled as an external impulse that prompts a distributed pattern of interictal epileptiform discharges

6. In Equation 1: in my opinion, the correct description of the relationship between t and p should be: the variable time t is independent of position p . I don't think that b_{uniform} defines a constant relationship between position and time as the authors claimed. A linear relationship can be constant, but that is not the case here.

Constant here refers to the 'constant function' where the output (predicted latency time t) is the same for every input (position p). If we wanted to write down t as a function of p , we would therefore want to write down $t = f(p) = b$; where b is a constant value, which does not change regardless of p .

We have added the following to try and clarify

If there is no spatiotemporal pattern, time of detection should be independent of channel position, modelled through a simple constant function (where b_{uniform} defines a relationship where t is independent of p , thus on average constant with respect to changes in position p):

Dear Dr Rosch,

Re: JP-RP-2025-288141R1 "Interictal discharges spread along local recurrent networks between tubers and surrounding cortex" by Stasa Tumpa, Rachel Thornton, Martin Tisdall, Torsten Baldeweg, Karl Friston, and Richard Ewald Rosch

Thank you for submitting your manuscript to The Journal of Physiology. It has been assessed by a Reviewing Editor and by 3 expert referees and we are pleased to tell you that it is acceptable for publication following satisfactory revision.

REVISION CHECKLIST:

We look forward to receiving your revised submission.

Yours sincerely,

Richard Carson
Senior Editor
The Journal of Physiology

REQUIRED ITEMS

- You must start the Methods section with a paragraph headed Ethical Approval. If experiments were conducted on humans, confirmation that informed consent was obtained, preferably in writing, that the studies conformed to the standards set by the latest revision of the Declaration of Helsinki and that the procedures were approved by a properly constituted ethics committee, which should be named, must be included in the article file. If the research study was registered (clause 35 of the Declaration of Helsinki), the registration database should be indicated, otherwise the lack of registration should be noted as an exception (e.g. The study conformed to the standards set by the Declaration of Helsinki, except for registration in a database). For further information see: <https://physoc.onlinelibrary.wiley.com/hub/human-experiments>.
- Your manuscript must include a complete Additional Information section, including competing interests; funding; author contributions and acknowledgements.
- A Data Availability Statement is required for all papers reporting original data. This must be in the Additional Information section of the manuscript itself. It must have the paragraph heading 'Data Availability Statement'. All data supporting the results in the paper must be either: in the paper itself; uploaded as Supporting Information for Online Publication; or archived in an appropriate public repository. The statement needs to describe the availability or the absence of shared data. Authors must include in their statement: a link to the repository they have used, or a statement that it is available as Supporting Information; reference the data in the appropriate sections(s) of their manuscript; and cite the data they have shared in the References section. Whenever possible, the scripts and other artefacts used to generate the analyses presented in the paper should also be publicly archived. If sharing data compromises ethical standards or legal requirements then authors are not expected to share it, but must note this in their statement. For more information, see our Statistics Policy.

EDITOR COMMENTS

Reviewing Editor:

Ethics Concerns:

Were the studies done in accordance with the Declaration of Helsinki and was informed consent obtained from the participants in written form?

Comments for Authors to ensure the paper complies with the Statistics Policy (Required):

Please use standard deviation instead of standard error of the mean.

Comments to the Author:

Thank you for submitting the revision of your article. All comments of the reviewers have been adequately addressed. Nevertheless, some minor points are still open:

- The authors may have overlooked my comments on the limitations section, which should also address the limitations of the modeling approach, in particular dynamic causal modeling.

- The author contributions are missing.
- The data availability statement is missing (only a short statement on some part of the code is present).

Please also see 'Required Items' above.

REFeree COMMENTS

Referee #1:

My comments have been adequately addressed.

Referee #2:

I do not have any further comments.

Referee #3:

Thank you for submitting your manuscript to The Journal of Physiology.

It is a requirement to start the Methods section with the sub-heading "Ethical Approval".

Thank you for providing detail on the approvals that were in place for this study, including the specific approval codes from the national regulator and local approval body.

It is evident that the study is a retrospective analysis of data gathered during clinical assessments of patients.

Can you please confirm the following: Approvals in hand explicitly provide for a waiver of the requirement for patient (parental/legal guardian) consent, which presumably was granted on the basis that all procedures were part of standard clinical practice and not shaped a priori by a research question that might necessitate a procedure, in whole or part, informed by the research question independent of standard clinical practice.

It is our understanding based on the submitted article that these were the conditions for the approvals granted, which would in turn waive The Journal's requirement for a statement in the manuscript relating to written informed consent and a statement confirming adherence to the principles of the Declaration of Helsinki, which are otherwise, in the context of a research study involving human participants, required.

Please confirm that the approvals provide for a waiver of consent.

END OF COMMENTS

REQUIRED ITEMS

You must start the Methods section with a paragraph headed Ethical Approval. If experiments were conducted on humans, confirmation that informed consent was obtained, preferably in writing, that the studies conformed to the standards set by the latest revision of the Declaration of Helsinki and that the procedures were approved by a properly constituted ethics committee, which should be named, must be included in the article file. If the research study was registered (clause 35 of the Declaration of Helsinki), the registration database should be indicated, otherwise the lack of registration should be noted as an exception (e.g. The study conformed to the standards set by the Declaration of Helsinki, except for registration in a database). For further information see: <https://physoc.onlinelibrary.wiley.com/hub/human-experiments>.

We have now addressed this in the comments to referee 3 below, by adding the following detailed section to the Methods:

2.1 Ethical Approval

Anonymised data were collected retrospectively from recordings performed according to standard clinical practice for a clinical indication. The study did not influence clinical decision making, and the approach to recording intracranial EEG was not altered by the research question; all care for the patients was provided according to standard clinical practice. The retrospective use of anonymised clinical data was approved by the UK Health Regulatory Authority (HRA, IRAS ID 229772) and the Great Ormond Street Hospital / UCL Great Ormond Street Institute of Child Health Joint Research Office (Project ID 17NP05). HRA approval specifically waived the requirement for written consent for this retrospective analysis of anonymised data.

- Your manuscript must include a complete Additional Information section, including competing interests; funding; author contributions and acknowledgements.

We have now added these as follows

- A Data Availability Statement is required for all papers reporting original data. This must be in the Additional Information section of the manuscript itself. It must have the paragraph heading 'Data Availability Statement'. All data supporting the results in the paper must be either: in the paper itself; uploaded as Supporting Information for Online Publication; or archived in an appropriate public repository. The statement needs to describe the availability or the absence of shared data. Authors must include in their statement: a link to the repository they have used, or a statement that it is available as Supporting Information; reference the data in the appropriate section(s) of their manuscript; and cite the data they have shared in the References section. Whenever possible, the scripts and other artefacts used to generate the analyses presented in the paper should also be publicly archived. If sharing data compromises ethical standards or legal requirements then authors are not expected to share it, but must note this in their statement. For more information, see our Statistics Policy.

This has now been added as follows

5.6 Data Availability Statement

Where possible without compromising anonymity of the patients, data are available upon reasonable request. This may include, e.g. epoched recordings of interictal epileptiform discharge groups with contacts labelled in reference to the lesion. All custom code can be accessed freely online (https://github.com/roschkoenig/Travelling_Spikes).

EDITOR COMMENTS

Reviewing Editor:

Ethics Concerns:

Were the studies done in accordance with the Declaration of Helsinki and was informed consent obtained from the participants in written form?

The requirement for informed consent was waived by the UK Health Regulatory Authority in view of anonymous collection of data, acquired for a clinical indication, see response to reviewer below. As such the Declaration of Helsinki does not apply as patients received standard clinical care and were not subjected to any change to their care based on the research question.

Comments for Authors to ensure the paper complies with the Statistics Policy (Required):
Please use standard deviation instead of standard error of the mean.

This has now been changed in Figure 3 and its Figure legend

Comments to the Author:

Thank you for submitting the revision of your article. All comments of the reviewers have been adequately addressed. Nevertheless, some minor points are still open:
The authors may have overlooked my comments on the limitations section, which should also address the limitations of the modeling approach, in particular dynamic causal modeling.

We have now added the following to the limitations section

4.4 Limitations

There are several important limitations to the study. We are fundamentally constrained by the data that is available – although SEEG allows a unique insight into in vivo dynamics of the epileptic brain, the recordings are indirect, macroscopic summaries of a large number of individual neurons. Inferring synaptic parameters at this macroscale may therefore be limited. **Dynamic causal modelling offers one approach to address this ill-posed problem - i.e. inferring synaptic parameters from summary statistics of neuronal activity, like the EEG signal. However, due to computational constraints this inference relies on expectation-maximisation and local gradient descent - meaning that it does not always identify a globally optimal parameter, given a dataset, but may get stuck in local minima. Secondly, the modelling is dependent on choice of network structures included in the model comparison, and the choice of prior values. Future updates on e.g. optimal priors for parameters for intracranial EEG recording may therefore alter some of the inference presented here.**

- The author contributions are missing.
- The data availability statement is missing (only a short statement on some part of the code is present).

These both have now been included in the 'Additional Information' section as below

5 Additional Information

5.1 Competing Interests

None of the authors has a competing interest to disclose.

5.2 Funding

This work is supported by the National Institute for Health Research Biomedical Research Centre at Great Ormond Street Hospital for Children NHS Foundation Trust and University College London. The project was funded by the Oakgrove Charitable Foundation (MT, RT, ST, RER), and the Wellcome Trust (RER: 209164/Z/17/Z, 106556/Z/14/Z, KJF: 088130/Z/09/Z)

5.3 Author Contributions

- S Tumpa: Formal analysis, Investigation, Data curation, Writing – original draft
- R Thornton: Conceptualisation, Investigation, Data curation, Writing – review and editing, Funding acquisition
- M Tisdall: Conceptualisation, Investigation, Data curation, Writing – review and editing, Funding acquisition
- T Baldeweg: Conceptualisation, Writing – review and editing
- KJ Friston: Conceptualisation, Methodology, Writing – review and editing
- RE Rosch: Conceptualisation, Methodology; Formal analysis; Investigation, Writing – original draft, Writing – review and editing; Visualisation, Supervision, Funding acquisition

5.4 Acknowledgements

We are grateful to all the patients and their families, as well as to the clinical teams overlooking their care.

5.6 Data Availability Statement

Where possible without compromising anonymity of the patients, data are available upon reasonable request. This may include, e.g. epoched recordings of interictal epileptiform discharge groups with contacts labelled in reference to the lesion. All custom code can be accessed freely online (https://github.com/roschkoenig/Travelling_Spikes).

Please also see 'Required Items' above.

We have addressed these as above

REFeree COMMENTS

Referee #1:

My comments have been adequately addressed.

Thank you for the helpful feedback that helped us improve our paper.

Referee #2:

I do not have any further comments.

We appreciate your feedback and the opportunity to address them

Referee #3:

Thank you for submitting your manuscript to The Journal of Physiology.

It is a requirement to start the Methods section with the sub-heading "Ethical Approval".

This has now been added

Thank you for providing detail on the approvals that were in place for this study, including the specific approval codes from the national regulator and local approval body. It is evident that the study is a retrospective analysis of data gathered during clinical assessments of patients.

Can you please confirm the following: Approvals in hand explicitly provide for a waiver of the requirement for patient (parental/legal guardian) consent, which presumably was granted on the basis that all procedures were part of standard clinical practice and not shaped a priori by a research question that might necessitate a procedure, in whole or part, informed by the research question independent of standard clinical practice.

We have now confirmed this in the methods section of the paper as follows.

2.1 Ethical Approval

Anonymised data were collected retrospectively from recordings performed according to standard clinical practice for a clinical indication. The study did not influence clinical decision making, and the approach to recording intracranial EEG was not altered by the research question; all care for the patients was provided according to standard clinical practice. The retrospective use of anonymised clinical data was approved by the UK Health Regulatory Authority (HRA, IRAS ID 229772) and the Great Ormond Street Hospital / UCL Great Ormond Street Institute of Child Health Joint Research Office (Project ID 17NP05). HRA approval specifically waived the requirement for written consent for this retrospective analysis of anonymised data.

It is our understanding based on the submitted article that these were the conditions for the approvals granted, which would in turn waive The Journal's requirement for a statement in the manuscript relating to written informed consent and a statement confirming adherence to the principles of the Declaration of Helsinki, which are otherwise, in the context of a research study involving human participants, required.

Thank you - we hope that the added clarification in the manuscript is sufficient to highlight that this is a retrospective analysis of standard clinical data and that the clinical care for the patients was not altered by the study question or analysis.

Please confirm that the approvals provide for a waiver of consent.

We have now included this as follows

HRA approval specifically waived the requirement for written consent for this retrospective analysis of anonymised data.

Dear Dr Rosch,

Re: JP-RP-2025-288141R2 "Interictal discharges spread along local recurrent networks between tubers and surrounding cortex" by Stasa Tumpa, Rachel Thornton, Martin Tisdall, Torsten Baldeweg, Karl Friston, and Richard Ewald Rosch

We are pleased to tell you that your paper has been accepted for publication in The Journal of Physiology.

Yours sincerely,

Richard Carson
Senior Editor
The Journal of Physiology

If you would like to receive our 'Research Roundup', a monthly newsletter highlighting the cutting-edge research published in The Physiological Society's family of journals (The Journal of Physiology, Experimental Physiology, Physiological Reports, The Journal of Nutritional Physiology and The Journal of Precision Medicine: Health and Disease), please click this link, fill in your name and email address and select 'Research Roundup':

<https://www.physoc.org/journals-and-media/membernews>

- **TRANSPARENT PEER REVIEW POLICY:** To improve the transparency of its peer review process, The Journal of Physiology publishes online as supporting information the peer review history of all articles accepted for publication. Readers will have access to decision letters, including Editors' comments and referee reports, for each version of the manuscript as well as any author responses to peer review comments. Referees can decide whether or not they wish to be named on the peer review history document.
- You can help your research get the attention it deserves! Check out Wiley's free Promotion Guide for best-practice recommendations for promoting your work at: www.wileyauthors.com/eeo/guide. You can learn more about Wiley Editing Services which offers professional video, design, and writing services to create shareable video abstracts, infographics, conference posters, lay summaries, and research news stories for your research at: www.wileyauthors.com/eeo/promotion.
- **IMPORTANT NOTICE ABOUT OPEN ACCESS:** To assist authors whose funding agencies mandate public access to published research findings sooner than 12 months after publication, The Journal of Physiology allows authors to pay an Open Access (OA) fee to have their papers made freely available immediately on publication.

EDITOR COMMENTS

Reviewing Editor:

Many thanks for submitting your revision. The remaining points on ethics, the limitations section, and the author contributions have been solved. A minor point is still remaining on the data availability, which can be addressed at proof stage: Please provide the anonymized data supporting the results in an online repository, according to the J Physiol guidelines:

"[...] All data supporting the results in the paper must be either: in the paper itself; uploaded as Supporting Information for Online Publication; or archived in an appropriate public repository. [...]"